# Phase Change Process in a Zigzag Plate Latent Heat Storage System during Melting and Solidification

**DOI:** 10.3390/molecules25204643

**Published:** 2020-10-12

**Authors:** Roohollah Babaei Mahani, Hayder I. Mohammed, Jasim M. Mahdi, Farhad Alamshahi, Mohammad Ghalambaz, Pouyan Talebizadehsardari, Wahiba Yaïci

**Affiliations:** 1Institute of Research and Development, Duy Tan University, Da Nang 550000, Vietnam; roohollahbabaeimahani@duytan.edu.vn; 2Faculty of Civil Engineering, Duy Tan University, Da Nang 550000, Vietnam; 3Department of Physics, College of Education, University of Garmian, Kurdistan 46021, Iraq; hayder.i.mohammad@garmian.edu.krd; 4Department of Energy Engineering, University of Baghdad, Baghdad 10071, Iraq; jasim@siu.edu; 5Department of Mechanical Engineering, Amirkabir University of Technology, Tehran 1591634311, Iran; farhad.alamshahi@gmail.com; 6Metamaterials for Mechanical, Biomechanical and Multiphysical Applications Research Group, Ton Duc Thang University, Ho Chi Minh City 758307, Vietnam; mohammad.ghalambaz@tdtu.edu.vn; 7Faculty of Applied Sciences, Ton Duc Thang University, Ho Chi Minh City 758307, Vietnam; 8CanmetENERGY Research Centre, Natural Resources Canada, 1 Haanel Drive, Ottawa, ON K1A 1M1, Canada

**Keywords:** phase change material, melting, solidification, corrugated plate heat exchanger, latent heat thermal energy storage

## Abstract

Applying a well-performing heat exchanger is an efficient way to fortify the relatively low thermal response of phase-change materials (PCMs), which have broad application prospects in the fields of thermal management and energy storage. In this study, an improved PCM melting and solidification in corrugated (zigzag) plate heat exchanger are numerically examined compared with smooth (flat) plate heat exchanger in both horizontal and vertical positions. The effects of the channel width (0.5 W, W, and 2 W) and the airflow temperature (318 K, 323 K, and 328 K) are exclusively studied and reported. The results reveal the much better performance of the horizontal corrugated configuration compared with the smooth channel during both melting and solidification modes. It is found that the melting rate is about 8% faster, and the average temperature is 4 K higher in the corrugated region compared with the smooth region because of the large heat-exchange surface area, which facilitates higher rates of heat transfer into the PCM channel. In addition to the higher performance, a more compact unit can be achieved using the corrugated system. Moreover, applying the half-width PCM channel accelerates the melting rate by eight times compared to the double-width channel. Meanwhile, applying thicker channels provides faster solidification rates. The melting rate is proportional to the airflow temperature. The PCM melts within 274 s when the airflow temperature is 328 K. However, the melting time increases to 460 s for the airflow temperature of 308 K. Moreover, the PCM solidifies in 250 s and 405 s in the cases of 318 K and 328 K airflow temperatures, respectively.

## 1. Introduction

Targets to meet the perpetual human desire for abundant on-demand energy lead the global energy consumption to grow at a faster rate. To live a life of comfort and ease, humans need energy to drive machines that do or help to do many activities related to human daily life like cooking, air-conditioning, food preservation, washing clothes, planting and harvesting, etc. [1]. An important consideration to improve the energy efficiency of these machines is by shifting some of the peak energy demand to the off-peak demand period by employing thermal energy storage (TES) technology [2]. One approach to achieve TES is by changing the temperature of the storage material by adding or removing heat while the material preserves the phase as liquid (like water, oil-based liquids, molten salts, etc.) or solid (like rocks, metals, and others) which is called sensible TES. Another approach to achieve TES is by changing the phase of the storage material from solid to liquid or vice versa which is called latent heat TES and the storage materials used are called phase-change materials, or PCMs. This approach is advantageous as the melting and solidification happen at an almost constant or nearly constant temperature, which is the phase-transition temperature [3]. This is quite favorable for applications requiring strict operation temperatures. Furthermore, comparing PCM-based TES to other methods, it has a higher storage density, thus allowing a more compact design of the storage component. Therefore, using latent heat TES components typically provides a more compact system with higher storage density than using sensible TES components. 

One of the main human activities contributing to high energy consumption, especially during the summer season is air-conditioning (AC) [4]. According to the International Energy Agency (IEA) [5], the use of air-conditioners and electric fans for cooling is responsible for about 20% of all electricity consumption in buildings worldwide. The concept of incorporating PCMs into AC systems has been widely explored in the open literature [6,7,8,9,10]. The aim was to attempt wholly or partially shifting of the electricity load demand from peak (high cost-tariff) times to off-peak (lower cost-tariff) periods. PCM phase transformation provides an effective approach to reduce AC electricity consumption during daytime by utilizing the stored cold energy during the night. In this approach, the cold ambient temperature at night is used by storing the cold thermal energy for later use in the daytime. This is quite beneficial in reducing the condenser temperature of the AC system leading to a higher efficiency. Yamaha et al. [11] examined the use of different paraffin mixtures packed in air ducts as a TES component for air-conditioning equipment of an ordinary office building in Nagoya (Japan). The findings indicated that a ratio of 5.4 kg PCM/m^2^ is enough to preserve almost a constant room temperature without any cold source operation from 1:00 to 4:00 p.m. Zhao et al. [12] proposed a PCM-based shell-and-tube TES device integrated with a conventional air-conditioner to improve cooling performance. Water and air were used as the heat transfer fluids (HTF) in charging and discharging loops, respectively. The study reported a 25.6% increase in the proposed system’s coefficient of performance (COP) compared with the conventional water-cooled AC system. Chaiyat and Kiatsiriroat [13] developed a simulation model to evaluate the cooling efficiency of an AC system modified with paraffin-based PCM under the Thai climate. It was found that the modified system saved around 9.10% of the electricity cost with a payback period of 4.15 years. Rahdar et al. [14] studied the performance of AC equipment with PCM and an ice TES tank in office buildings. The study showed that power consumption could be saved by 4.59% and 7.58% and CO_2_ emission could be reduced by 17.8% and 27.2% with the integration of ice and PCM, respectively. Aljehani et al. [15] proposed a conceptual design of the AC system integrated with a composite TES component consisting of PCM and expanded graphite. The study reported 30% lower power consumption by the compressor with 30% lower CO_2_ emission production compared with a conventional AC system. Recently, Nie et al. [16] tested the cooling performance of an AC system with PCM for transport applications. Findings showed an increase in the COP by 19.1%, with an overall reduction in the power consumption cost by 17.8% compared with the conventional AC system. 

The phase change model could be involved with natural convection effects in the molten region, as discussed in [17,18]. Various aspects of natural convection heat transfer in enclosures have been investigated such as Magnetohydrodynamic effects [19,20], entropy generation [21,22], and nanofluids [23]. Despite some degree of natural convection heat transfer, which could improve the melting heat transfer in thermal energy storage containers, the low heat transfer performance of PCMs is a barrier for quick charging and discharging processes. 

As mentioned, one of the key challenges that limits the effective exploitation of PCMs is their poor thermal conductivities which mostly fall into the range of 0.1 W/(m K) to 0.8 W/(m K) [4]. Such low values delay the energy charging/discharging response rates, particularly during solidification mode, which greatly impacts the dynamics of the AC system’s cooling performance at peak load hours. Therefore, several heat-transfer enhancement techniques have been introduced and applied to improve the thermal response of PCM-based TES systems [24,25]. These techniques include incorporating fins and extended surfaces [26,27,28,29], modifying the geometry [30,31,32], embedding porous structures [33,34,35,36,37], composing thermally high conducting particles [38,39,40,41], using multiple PCMs [42,43], nano-encapsulation [44,45,46], and using the combinations of different methods [43,47]. Among them, incorporating with fins is a particularly attractive option from both economic and engineering perspectives. This is due to their high thermal efficacy, ease and simplicity in design, and relatively low-cost fabrication [48]. The PCMs are typically enclosed in containers to avoid leakage. Hence, the geometry of containers, placement of fins or metal foam could notably influence the thermal performance of TES.

Increasing the surface area for heat transfer via fins or geometry itself enhances the melting and solidification rates of the PCM system [49,50,51]. Sathe and Dhoble [52] analyzed the dynamic thermal efficiency of PCM charging in an inclined finned rectangular enclosure with a top surface heating source. They found that the charging rate is not significantly affected by combining extra fins (more than three), although it considerably enhances the PCM thermal efficiency. Yeldiz et al [50] numerically studied the thermal convection inside a PCM enclosure combined with two types of fins including rectangular and tree-shape branched fins. They found that the rectangular fin is more effective than the tree-shape fin with the same mass. Yu et al. [51] examined the potential of a fractal tree-like fin to improve the melting property of PCM in latent heat TES units. It was found that the good distribution of tree-like fins can help to develop the melting process due to the more uniform temperature nature of tree-like fins compared with that of the corresponding plate fins.

Interruptions in the heat-exchange surface like a zigzag configuration can bring significant improvement in the melting and solidification rates, thereby contributing to a superior storage performance of the PCM-based TES system. This type of surface interruption is preferable since the zigzag configuration enables the PCM in charge to have a larger heat-exchange surface area compared with the conventional flat-plate heat exchanger. This makes the heat energy absorption and release higher, thus the melting and solidification of PCM proceed at faster rates. According to the literature, very few studies up to now have been conducted to explore the use of a zigzag-plate heat exchanger as a high performance design for PCM containment. Wang et al. [53] investigated the effect of multiple-PCM arrangement within a zigzag plate TES unit. They recognized that a potential melting improvement can be achieved if a larger temperature difference between the PCM layers is applied. In another work [54], they studied the solidification of multiple PCMs within a zigzag plate type containment. They stated that the zigzag containment brings a very positive impact on the temperature distribution and the overall solidification response of the multiple-PCM module.

In this study, the energy storage potential and thermofluidic behavior of a PCM within a zigzag plate TES system are studied. The effects of different widths of the corrugated surface and different temperatures of the heat transfer fluid are examined. Therefore, the main objective is to explore through numerical simulation the melting and solidification mechanisms of PCM in a zigzag-plate heat exchanger applicable to use with compact air-conditioners. Different zigzag-plate configurations are investigated to determine the optimized configuration under different heat load conditions. Different orientations of the heat exchanger including the horizontal and vertical arrangements are considered. The effects of different parameters are studied in terms of the temperature distribution, liquid-fraction contours, average-temperature curves, and liquid-fraction profile. Findings based on this study would contribute to the ongoing improvement efforts in the design, analysis and operation of TES systems for realistic applications. 

## 2. Problem Description

Figure 1 schematically represents the cross-section of the zigzag plate latent heat energy exchanger with one layer proposed in this study. Two straight channels with the same width of the zigzag section are introduced at the beginning and the end of the heat exchanger. 

The whole system consists of several layers of the zigzag plate (shown in Figure 1) where the PCM is located between two tubes of the heat-transfer fluid (HTF) (air) with the same geometry. Figure 2 displays the schematic of a PCM layer in the middle of two HTF tubes. The main focus of this study is on the effects of the corrugated plate heat exchanger during both melting and solidification compared with the straight channel for different dimensions of the heat exchanger. Therefore, it is assumed that the flow rate of the HTF is moderate enough to have an almost constant temperature at the walls of the PCM section and, as a result, a constant temperature boundary condition is used for the walls without modelling the airflow. In other words, it was assumed that the channel is isothermally cooled or warmed by the HTF flowing at constant temperature. This also enables the heat to only transfer to or from the PCM to realize only the effects of the corrugated plate on the phase change process. Furthermore, the effects of the HTF temperature were investigated comprehensively as the only effective parameters of the HTF. For the effect of the HTF Reynolds number, it was proved that by increasing the Reynolds number, the rate of heat transfer from the HTF to the PCM increases due to the higher convection heat transfer which has been widely established in the literature and therefore it is not investigated quantitatively in this study [55,56,57,58]. Furthermore, as mentioned, for high flow rates of the HTF, the assumption of considering constant wall temperature is meaningful [59] and increasing the velocity has a negligible effect as mentioned in different papers in the literature [42,48,59]. Note that the walls of the PCM layer on the right- and left-hand sides of the heat exchanger are considered insulated.

The geometrical details of the system are listed in Table 1. RT35 (Rubitherm) is employed as the PCM, which is among organic types of PCM. The properties of RT35 are presented in Table 2.

## 3. Mathematical Modelling

To reduce the simulation expenses and provide an accurate result with shorter operation time, some assumptions are applied to the model. The flow is assumed to be Newtonian, incompressible and laminar with a transient approach. The gravity is directed downward. Because of the low velocity, the viscous dissipation is neglected and the Boussinesq approximation applied to density and buoyant force. The outer wall of the tube is assumed to be well insulated with no heat loss to the ambient [61]. The velocity and temperature in the z-direction are constant and, therefore, the 2-dimensional geometry is assumed [42].

The conservation equations governed on the phase change problem, including the mass, momentum and energy (Equations (1)–(3), respectively) equations are given briefly as follows based on the enthalpy–porosity method introduced by Brent et al. [61]. This method utilized due to its ability to solve explicit tracking of the solid–liquid interface as well as providing a simple operation of phase-change problems [62,63]:(1)$$\frac{\partial \rho }{\partial t}+\nabla .\rho \vec{V}=0.\;$$
(2)$$\frac{\partial }{\partial t}\left(\rho \vec{\upsilon }\right)+\nabla .\left(\rho \vec{\upsilon }\vec{\upsilon }\right)=-\nabla P+\nabla .\left(\mu \nabla \vec{\upsilon }\right)+\rho g+S$$
(3)$$\frac{\partial }{\partial t}\left(\rho H\right)+\nabla .\left(\rho \vec{\upsilon }H\right)=\nabla .\left(k\nabla T\right)$$
where ρ, t, V, P, μ, g are density, time, velocity, pressure, viscosity and gravity acceleration, respectively. H is the total volumetric enthalpy, which is the sum of sensible enthalpy (*h*) and the latent heat (*L*) as follows:(4)H=h+fL
(5)$$h=h_{ref.}+\int_{T_{ref.}}^{T}C_{p}dt$$

In Equations (4) and (5), f refers to the liquid-fraction of the PCM in melting and solidification processes calculated as:(6)$$f=\{\begin{array}{ll} 0 & T\;<\;T_{Solidus}\;\\ \frac{T-T_{Solidus}}{T_{Liquidus}-T_{Solidus}} & T_{Solidus}<\;T\;<\;T_{Liquidus}\\ 1 & T\;>\;T_{Liquidus} \end{array}$$

In Equation (2), *S* is the momentum source term in the mushy zone based on the enthalpy–porosity approach, which takes the following form:(7)$$S=\frac{{\left(1-f\right)}^{2}}{\left(f^{3}+\varepsilon \right)}\left(\vec{\upsilon }-{\vec{\upsilon }}_{s}\right)A_{mush}.\;$$
where the term $A_{mush}$ is the mushy zone constant reflecting the mushy zone morphology that illustrates how the velocity is reduced when the PCM solidifies. The constant ε is a small number to prevent division by zero when *T_solid_* is greater than *T* [61,64,65]. The following parameters were utilized in the present study: $A_{mush}=10^{5},\;\varepsilon \;=\;0.001,{\;\mathrm{and}\;T}_{ref.}=298.15\;\left(K\right)$.

The density variation is defined as:(8)$$\rho =\rho_{0}\left(1-\beta \left(T-T_{0}\right)\right).\;$$
where β is the thermal expansion coefficient. By substituting Equations (7) and (8) in Equation (2), the momentum equation is derived as:(9)$$\frac{\partial }{\partial t}\left(\rho_{0}\vec{\upsilon }\right)+\nabla .\left(\rho_{0}\vec{\upsilon }\vec{\upsilon }\right)=-\nabla P+\nabla .\left(\mu \nabla \vec{\upsilon }\right)+\left(\rho -\rho_{0}\right)g+\frac{{\left(1-f\right)}^{2}}{\left(f^{3}+\varepsilon \right)}\left(\vec{\upsilon }-{\vec{\upsilon }}_{s}\right)A_{mush}$$

To reduce the value of ρ  from the buoyancy term in the momentum equation, the Boussinesq approximation is accurate as long as changes in actual density are small when $\beta \left(T-T_{0}\right))\leq 1$.

## 4. Numerical Modelling and Validation

The computational fluid dynamic (CFD) simulation and grid generation are performed using ANSYS workbench. Moreover, the QUICK scheme is used to calculate the diffusion fluxes and convection while the PRESTO scheme is used for the pressure correction equation. The convergence criteria are also considered 10^−6^ for all the governing equations. 

A series of numerical simulations with different cell numbers of 172,016, 267,002, and 419,903 proves how generously the mesh was generated to have a grid-independent solution. The variation of the liquid-fraction with different numbers of cells are presented in Figure 3a. As shown, the results for all grid resolutions are identical with the maximum difference of less than 1.5% [66]. Therefore, the mesh with the cell number of 172,016 is sufficient for the modelling. These results prove that the solution is mesh-independent and reliable enough to be verified against other studies’ results. A similar analysis was also performed on the size of the time step results selecting 0.25 s as the time-step size. Figure 3b displays the generated mesh in the bends of the selected grid after the mesh study.

The FLUENT code was verified using the investigation of Mat et al. [67] for both temperature and liquid-fraction which was performed numerically and experimentally. Mat et al. [67] studied a fined double-tube latent heat storage unit using RT58 as the PCM with constant wall temperature as the boundary condition. As shown in Figure 4, an excellent agreement is achieved in comparison with the data of Mat et al. [67] in this study. In the temperature profile, the maximum deviation between the present numerical study and experimental data is almost 9% while it is around 5% compared with the numerical study of Mat et al. [67]. For the liquid-fraction, the maximum deviation is 4% between the present numerical results and the numerical study of Mat et al. [67].

## 5. Results and Discussion

### 5.1. Effect of the Corrugated System Compared with the Smooth Case

To design an effective zigzag plate latent heat TES system for typical air-conditioning applications, the energy charging and discharging rates during both melting and solidification modes were analyzed and compared. Three different cases were considered for the plate TES system, namely: the horizontal corrugated plate, vertical corrugated plate, and horizontal non-corrugated plate. The transient melting evolution for the three studied cases is demonstrated in Figure 5. It can be observed that the horizontal corrugated plate presents the highest melting performance, which has a delicate difference compared with the vertical case. However, the difference is more noticeable compared with the horizontal flat channel. For the same channel width (w), the total melting time is about 315 s in the corrugate case against 343 s in the smooth plate case so that the time saving due to the use of the corrugate channel is about 8%. The reason is because of the larger heat transfer surface area of the airflow for the corrugated cases compared with the smooth channel. Thus, more heat is transferred allowing PCM to experience a faster phase transition as a consequence. The corresponding PCM temperature profile is shown in Figure 6. The temperature in the corrugated configurations is always higher than that for the smooth surface for any instant time. At 400 s, the mean temperature of the PCM is 320 K, 323 K and 324 K for the smooth case, corrugated horizontal case, and corrugated vertical case, respectively. This implies that a 4 K higher temperature can be achieved if the corrugated channel is applied. Meanwhile, the corrugated horizontal case shows a slightly higher performance than the vertical case which is then used for further analysis.

The PCM solidification rate for the case of the corrugated plate remains quicker than that for the smooth flat plate over the entire solidification process, as shown in Figure 7. This is due to the larger surface area of the corrugated surface configuration. For the same size of the PCM in both cases, the total solidification time is 280 s in the case of the corrugated plate compared to 303 s in the case of the smooth plate. Therefore, the time saving due to the use of the corrugate channel is about 7% during the solidification mode. Figure 8 presents the average temperature of the PCM inside the channel for both cases. The temperature in the corrugated system drops faster than the smooth surface. This behavior is due to the effective role of the natural convection heat transfer, which is greater in the case of the corrugated configuration because of the larger surface area of heat transfer.

The contours of the liquid-fraction and temperature development in different configurations are shown in Figure 9, Figure 10 and Figure 11. Figure 9 shows that within 350 s, all the PCM is melted and the average temperature at this time is 320 K. The phase-change process is faster, and the average temperature is higher in the corrugated region compared with that in the smooth region. This is caused by the larger heat exchanger surface area in the corrugated region, which helps to transfer more heat to the system. The PCM solidifies within 300 s (shown in Figure 10), and the average temperature drops to 299 K. Due to the natural convection effect and the heat exchanger surface area, the temperature in the corrugated area drops faster. Note that the flow of melted PCM during both modes of melting and solidification is buoyancy-driven and the velocity remains too low to form fluid turbulence or vortices. The flow velocity of melted PCM in all cases within the scope of this study is in the order of 10^−4^ m/s. 

In Figure 11, development of the temperature and the liquid-fraction for the vertical corrugated channel is illustrated. The PCM is melted totally within 400 s. The end of the left-hand side of the channel (bottom part of the channel) remains solid due to the regional effect of gravity. This is clear also from the temperature contours, which show the top side is relatively warmer than the bottom side. 

### 5.2. Effect of the Channel Width in the Corrugated System

The effects of the channel width are illustrated in Figure 12, Figure 13, Figure 14 and Figure 15) for both melting and solidification modes. The channel of 0.5 W is thinner than the other two cases, and thus holds a smaller amount of PCM as shown in Figure 12. This helps the PCM to completely melt within 94 s, faster than the reference width case (W = 0.9 cm) and the case of double width (2 W), which melt in 332 s and 776 s, respectively. This implies that reducing the zigzag plate width to 0.5 W can save 72 % melting time compared the reference case of W=0.9 cm. This behavior is due to transferring the same amount of heat through the same surface area to half the volume of the PCM. Figure 13 compares the mean PCM temperature of the thinner case (0.5 W) with the other two cases of the reference width (W = 0.9 cm) and the double width (2 W). For the same reasons mentioned above, the thinner case reaches a higher melting temperature within a shorter duration compared with that for the other two cases under consideration. 

The width of the channel also has an influence on the solidification rate due to natural convection. Figure 14 shows that the PCM in the thinner channel solidifies faster (82 s) than the thicker channels. This behavior is also illustrated in Figure 15, which shows that the PCM in the 0.5 W width channel reaches the equilibrium temperature.

Figure 16 shows the melting and solidification process in both zigzag and smooth configurations for the case of the 2 W width channel. The processes in the zigzag configuration have a faster rate because the increase in the size and the surface area of the storage unit. The melting process in the zigzag channel takes less than 760 s, while it takes 840 s for the smooth configuration. The solidification for both processes finished takes 1200 s, but with the zigzag configuration, the solidification is faster, taking 450 s, and then it became slower. 

The PCM needs a short time (100 s) to melt when the width is narrow (0.5 W) as the channel contains less PCM (shown in Figure 17). The figure also shows that the temperature rises quickly (100 s). With the wider channel (2 W), the system needs a longer time to melt completely (700 s) (shown in Figure 18). In the zigzag area, the PCM melts faster because of the higher surface area, which enhances the convection heat transfer in the system. This behavior is also clear from the temperature contours which show that the average PCM temperature reaches 309 K in 700s.

Figure 19 shows the solidification process in the 0.5 W system. The PCM solidifies completely within 100 s and the average temperature of the PCM reaches 294 K by the mentioned time. The natural convection supports a faster release of heat from the PCM, allowing it to solidify at a faster rate. Within the larger volume of the PCM with a fixed surface area (Figure 20), the system needs more time to solidify the PCM. Again, in the zigzag region, the PCM solidifies faster than the smooth area due to the heat exchanger surface area and phase change domain.

### 5.3. Effect of HTF Temperature

The effects of the airflow temperature on the melting and solidification processes are illustrated in Figure 21, Figure 22, Figure 23
and Figure 24. Figure 21 shows the development of the liquid-fraction for different airflow temperatures. The melting process rate is proportional to the air temperature. The total PCM melts within 260 s when the heat exchanger temperature is 328 K, which increases to 450 s when the temperature is 308 K. This behavior is due to the temperature difference of the heat exchanger (between the airflow and the PCM), which allows higher heat transfer to the PCM in the case of higher temperature of the airflow. The temperature of the PCM increases during the melting process as shown in Figure 22. The average temperature of the totally melted PCM reaches 326 K over 300 s, when the airflow temperature is 325K. This time extends (360s) when the airflow temperature is 323 K and the PCM temperature rises to 321 K.

Lower airflow temperature leads to faster rate of solidification since the mean temperature of the PCM is low with lower airflow temperature. Thus, the critical phase change temperature is reached faster as shown in Figure 23. All the PCM solidifies over 240 s and 360 s in the cases of 318 K and 328 K airflow temperatures, respectively. This behavior is clear also from Figure 24. The average PCM temperature drops to 285 K and 297 K within 240s for the airflow temperatures of 318 K and 328 K, respectively, and in the case of 320 K, the PCM temperature reaches 294 K. 

Figure 25 shows the liquid-fraction contours in both the melting and solidification processes at a specific time and channel width for different airflow temperatures. Higher airflow temperature causes faster melting rate because of more heat transfer to the PCM due to the higher temperature difference between the flowing air and the surface temperature. The lower airflow temperature during the charging mode causes melting with minimal temperature, which helps the PCM to solidify at a faster rate as shown in the solidification side of the figure.

## 6. Conclusions

A PCM-based thermal energy storage system with a zigzag configuration for use in compact air-conditioners was investigated in this work. A two-dimensional simulation model based on ANSYS FLUENT software was established to simulate the energy charging and discharging behaviors of the system. The model was validated via experimental and numerical studies showing excellent agreement. The effect of applying different orientations of the zigzag channel compared with the smooth channel was investigated. The effects of different widths of the zigzag channel, as well as various HTF (air) temperatures were also investigated and reported. Based on the results, it can be concluded that the system with zigzag configuration shows slightly higher performance over the system with the smooth configuration in the vertical orientation. However, much better improvement can be achieved using the zigzag configuration in comparison with the smooth configuration in the case of horizontal orientation. The total saving time due to the use of the corrugated channel is about 8% and 7% during the melting and solidifying modes, respectively. Moreover, the PCM melting and solidifying rates are faster in the thinner channel as the system holds less PCM mass. It was found that reducing the zigzag plate width from 2 W (W = 0.9 cm) to W and 0.5 W can improve the melting rate by 8 and 12 times compared to the 2 W case, respectively. It is was also found that the melting rate is directly proportional to the airflow temperature. The total PCM melts within 274 s when the air temperature is 328 K. However, this time increases to 460 s when the temperature is 308 K. This is attributed to the fact that the PCM stores less heat at lower air temperatures. Moreover, the PCM solidifies in 250 s and 405 s in the cases of 318 K and 328 K airflow temperatures, respectively. The results of this study provide an insight into the possible optimization and improvement of the PCM containment design using a zigzag configuration to provide better control of the system size and the traditional thermal energy losses.

In the future studies, the authors are going to investigate the number of zigzags by changing the angles of the corrugated plate whilst maintaining a constant length. The HTF is going to be modelled completely and the effect of different configurations of the plate heat exchanger is going to be investigated as future developments.

## Figures and Tables

**Figure 1 molecules-25-04643-f001:**
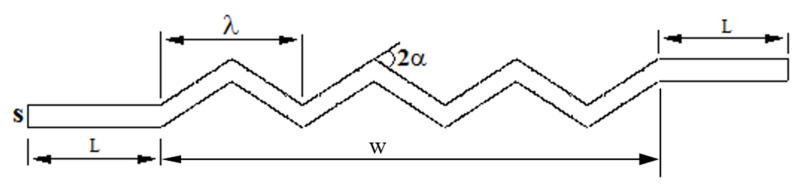
The geometry of the corrugated phase-change material (PCM)–air heat exchanger (one PCM domain).

**Figure 2 molecules-25-04643-f002:**
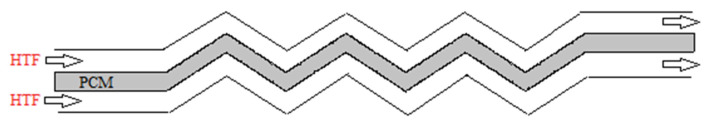
The schematic of the proposed system including the PCM layer and heat transfer fluids (HTF) (air).

**Figure 3 molecules-25-04643-f003:**
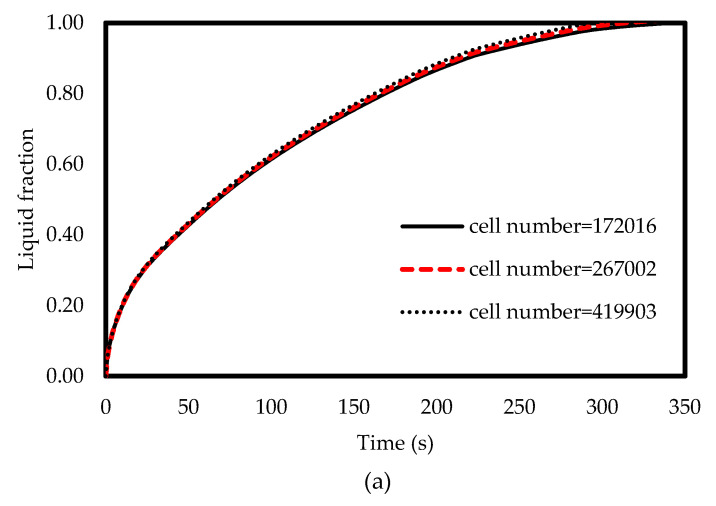
(**a**) Effect of grid resolution on liquid-fraction and (**b**) the selected grid.

**Figure 4 molecules-25-04643-f004:**
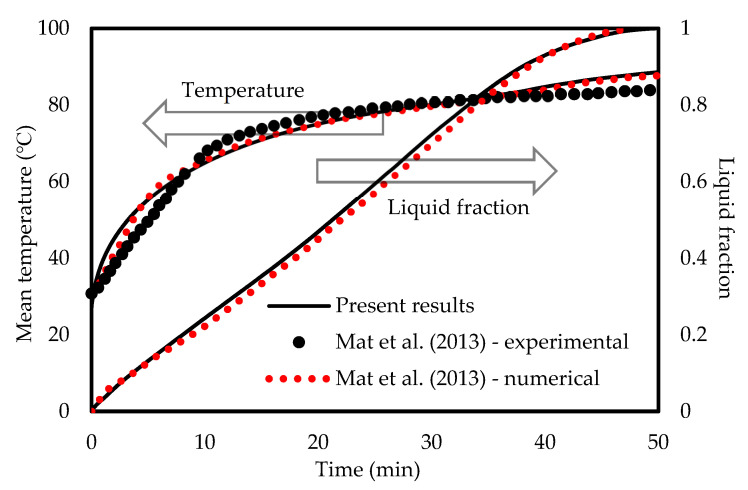
Code verification comparing with the data of Mat et al. [67].

**Figure 5 molecules-25-04643-f005:**
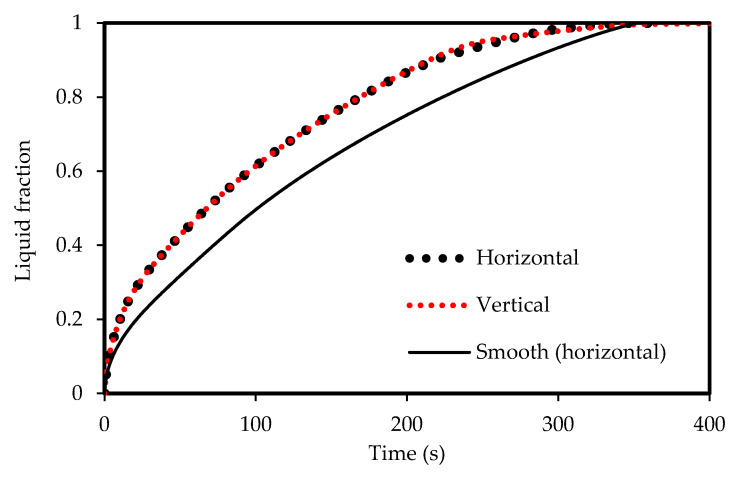
Liquid-fraction evolution during melting in the horizontal and vertical corrugated channel compared with smooth horizontal channel.

**Figure 6 molecules-25-04643-f006:**
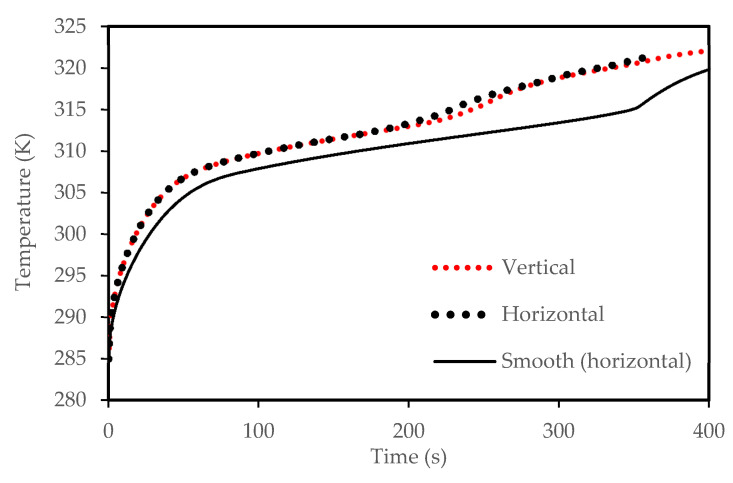
Temperature development in the horizontal and vertical corrugated channel compared with the smooth horizontal channel for the melting mode.

**Figure 7 molecules-25-04643-f007:**
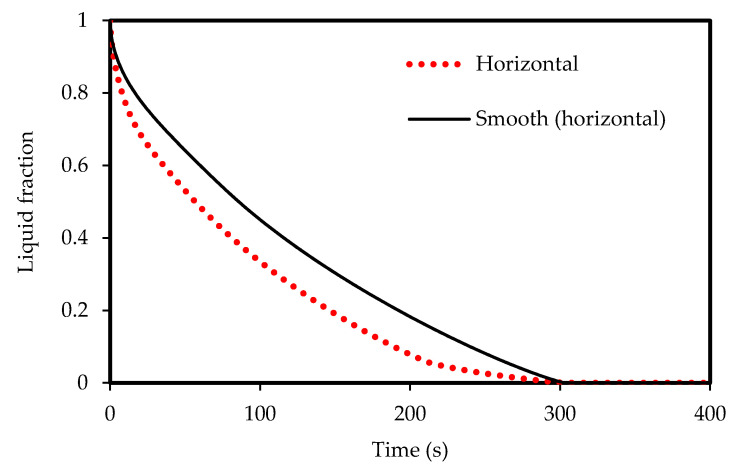
Liquid-fraction evolution during solidification in the horizontal corrugated channel and smooth horizontal channel.

**Figure 8 molecules-25-04643-f008:**
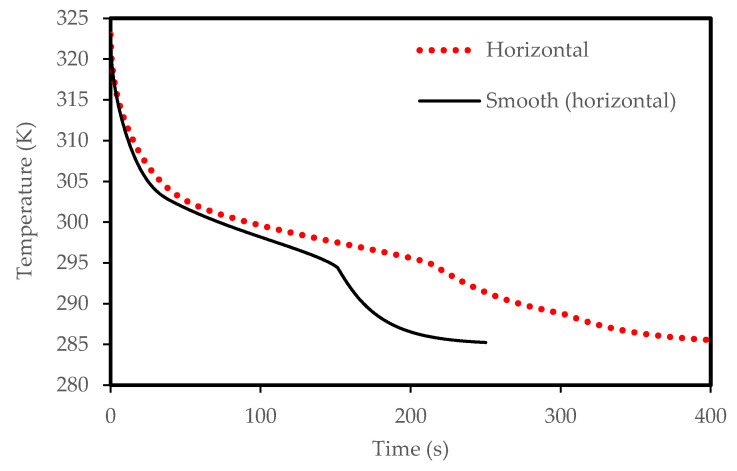
Temperature development in the horizontal corrugated channel and smooth horizontal channel for the solidification process.

**Figure 9 molecules-25-04643-f009:**
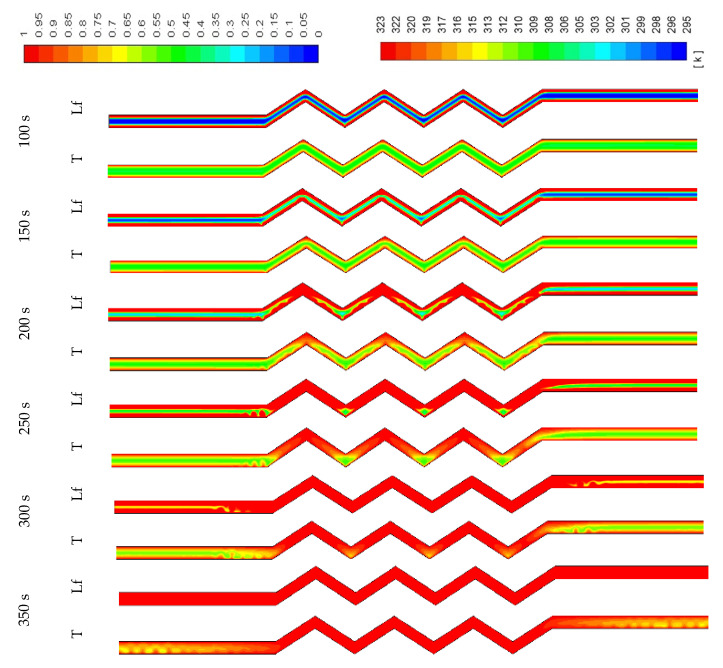
Contours of the liquid-fraction and temperature during the melting process in the horizontal corrugated channel for the standard width (0.9 cm).

**Figure 10 molecules-25-04643-f010:**
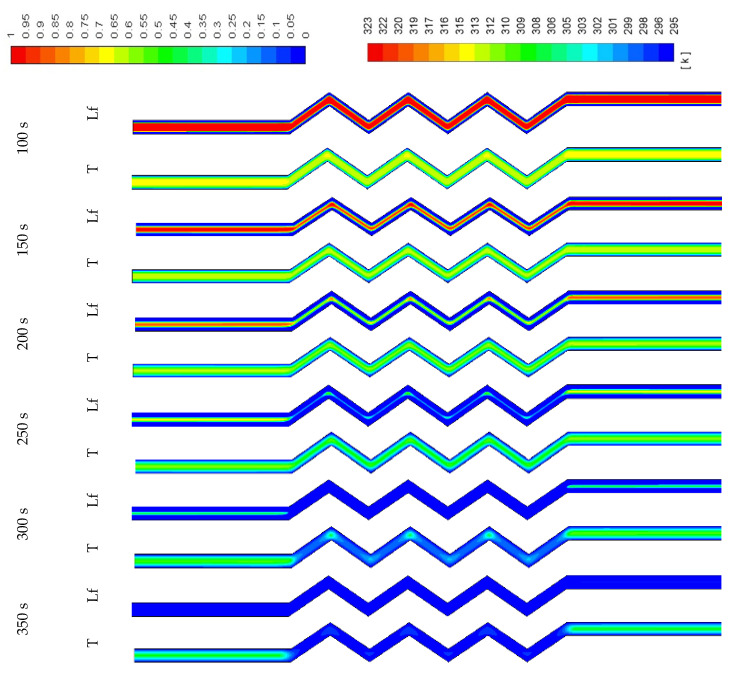
Contours of the liquid-fraction and temperature during the solidification process in the horizontal corrugated channel for the standard width (0.9 cm).

**Figure 11 molecules-25-04643-f011:**
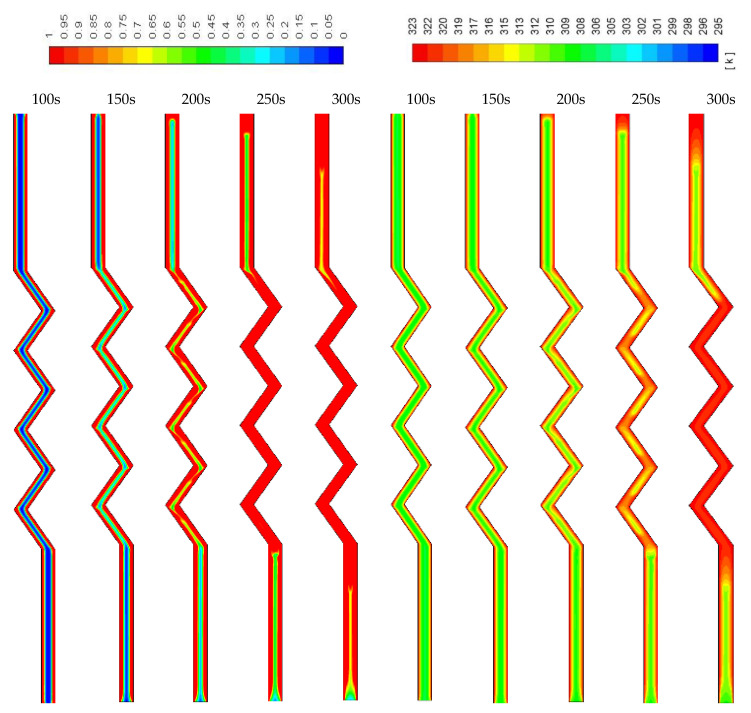
Contours of the liquid-fraction and temperature during the melting process for the standard width (0.9 cm) in the vertical corrugated channel.

**Figure 12 molecules-25-04643-f012:**
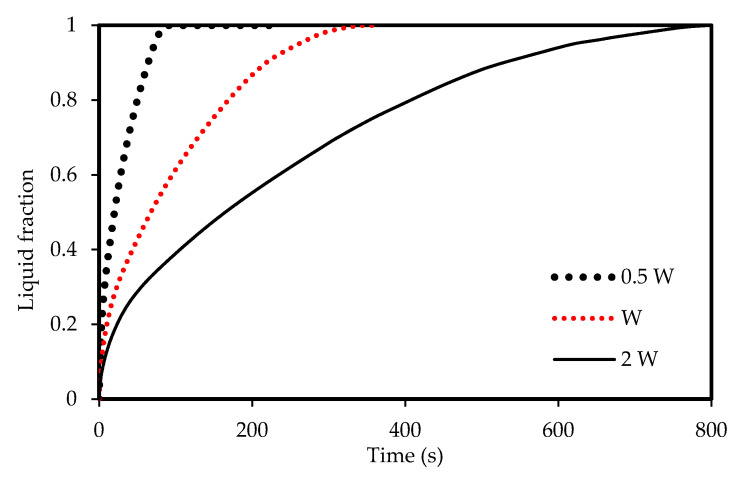
Evolution of the liquid-fraction for the melting process in the horizontal corrugated channel for different widths.

**Figure 13 molecules-25-04643-f013:**
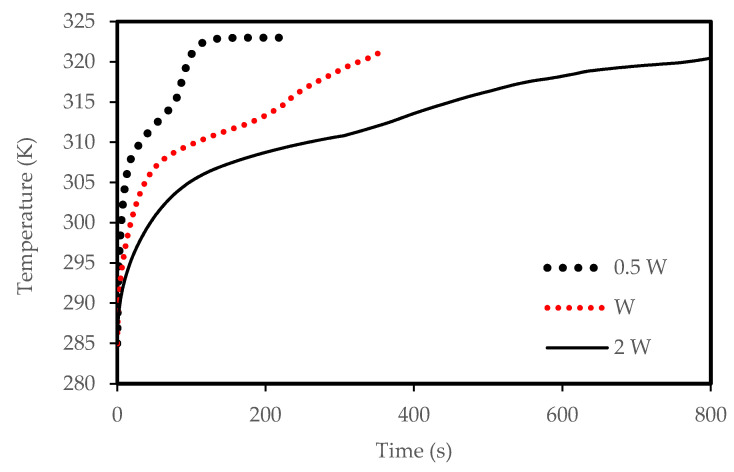
Temperature development in the horizontal corrugated channel for different widths for the melting process.

**Figure 14 molecules-25-04643-f014:**
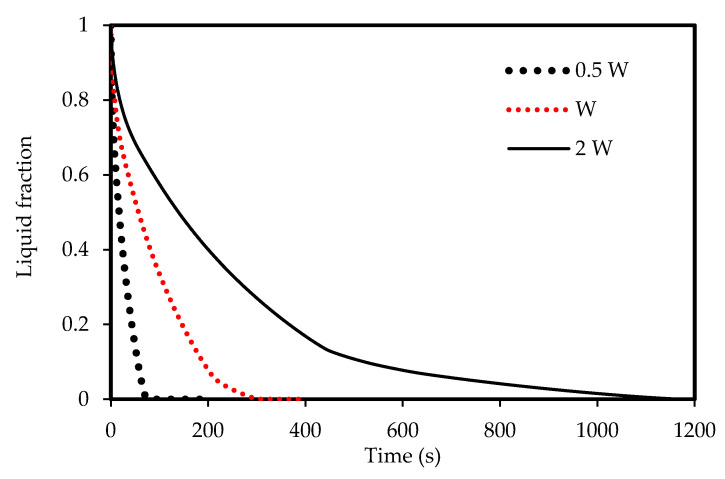
The liquid-fraction evolution of the solidification process in the horizontal corrugated channel for different widths.

**Figure 15 molecules-25-04643-f015:**
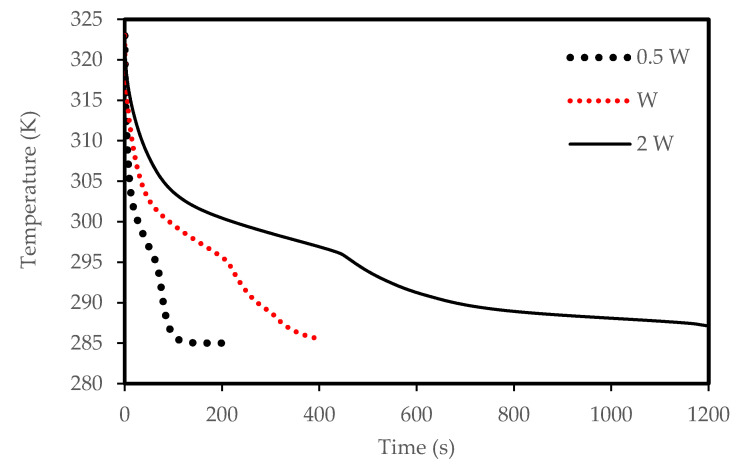
Temperature development in the horizontal corrugated channel for different widths for the solidification process.

**Figure 16 molecules-25-04643-f016:**
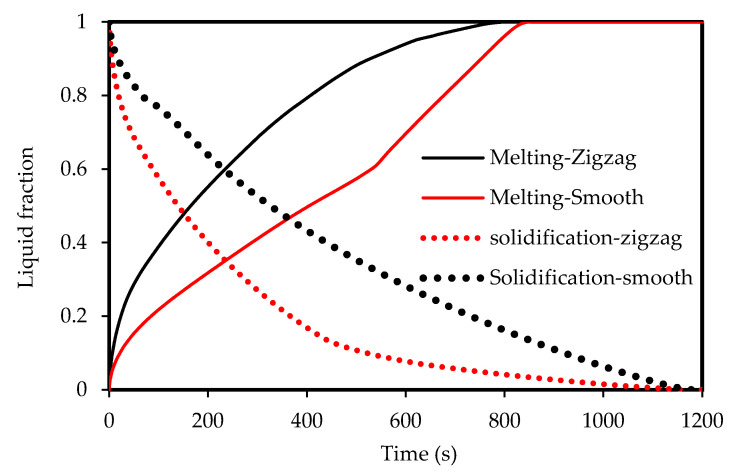
Evolution of the liquid-fraction in both melting and solidification processes for both Zigzag and smooth channels for the case 2 W width channel.

**Figure 17 molecules-25-04643-f017:**
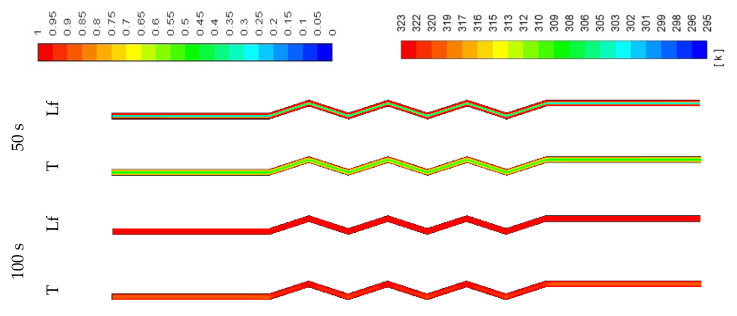
Contours of the liquid-fraction and temperature in the melting process for the half-width (0.5 W) horizontal corrugated channel.

**Figure 18 molecules-25-04643-f018:**
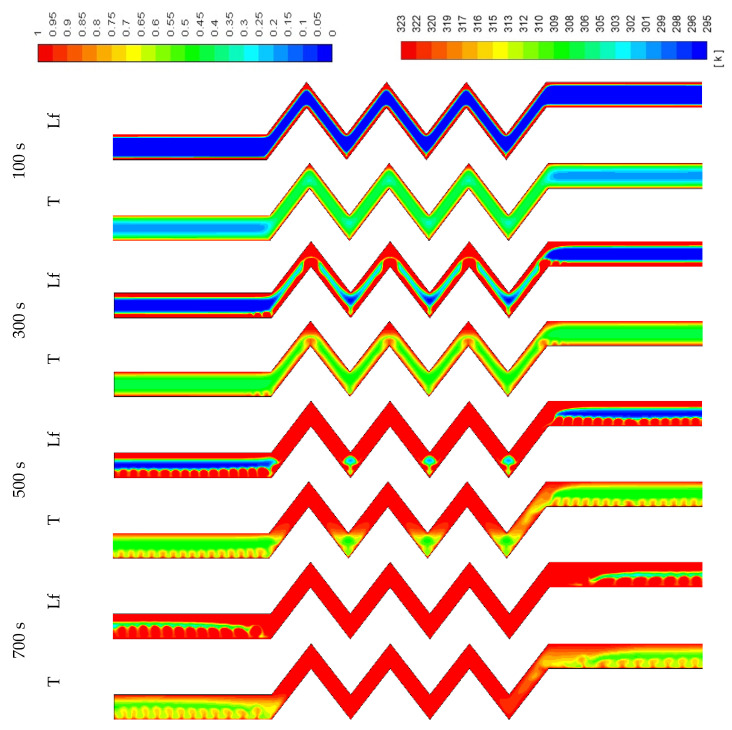
Contours of the liquid-fraction and temperature in a melting process for the double width (2 W) horizontal corrugated channel.

**Figure 19 molecules-25-04643-f019:**
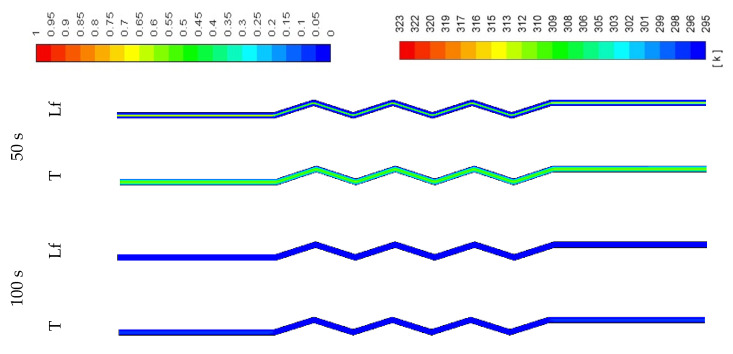
Contours of the liquid-fraction and temperature in a solidification process for the half-width (0.5 W) horizontal corrugated channel.

**Figure 20 molecules-25-04643-f020:**
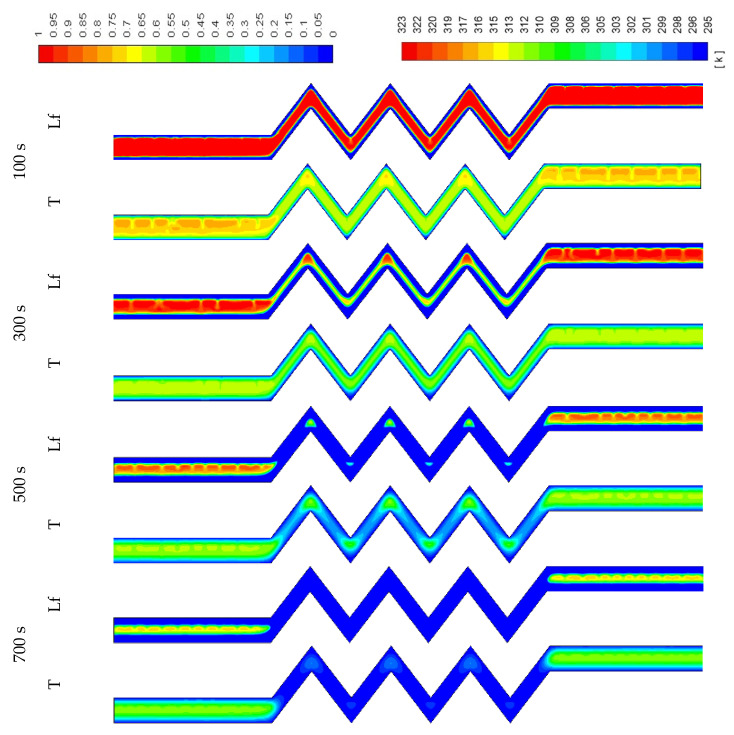
Contours of the liquid-fraction and temperature in a solidification process for the double width (2 W) horizontal corrugated channel.

**Figure 21 molecules-25-04643-f021:**
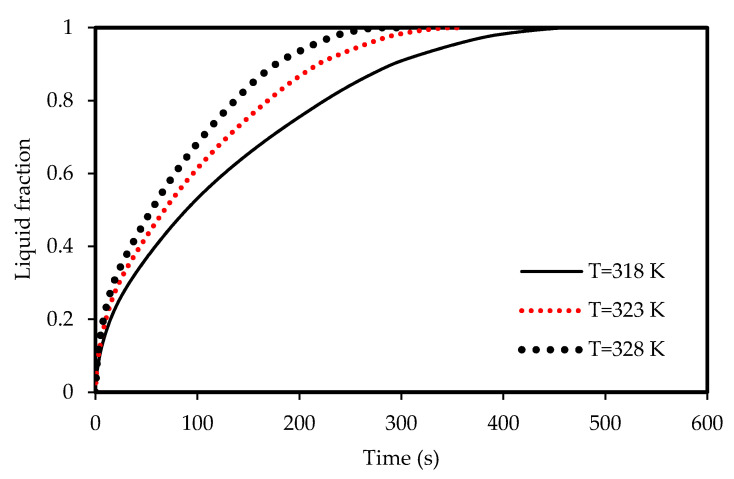
Evolution of the liquid-fraction of the melting process in the horizontal corrugated channel for different airflow temperatures.

**Figure 22 molecules-25-04643-f022:**
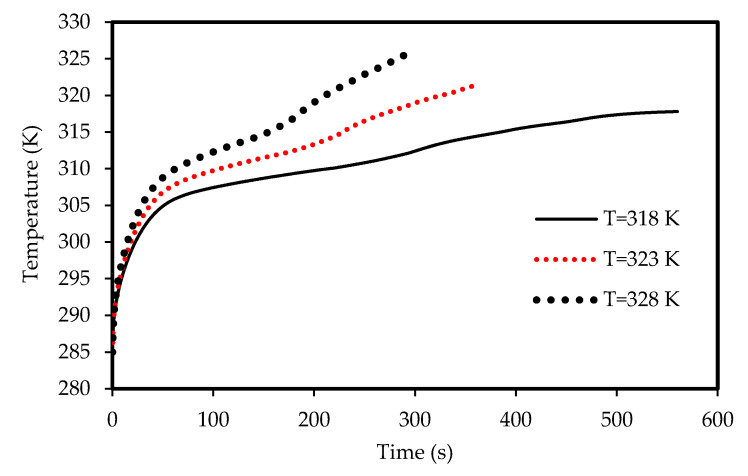
Variation of the temperatures for the melting process in the horizontal corrugated channel for different airflow temperatures.

**Figure 23 molecules-25-04643-f023:**
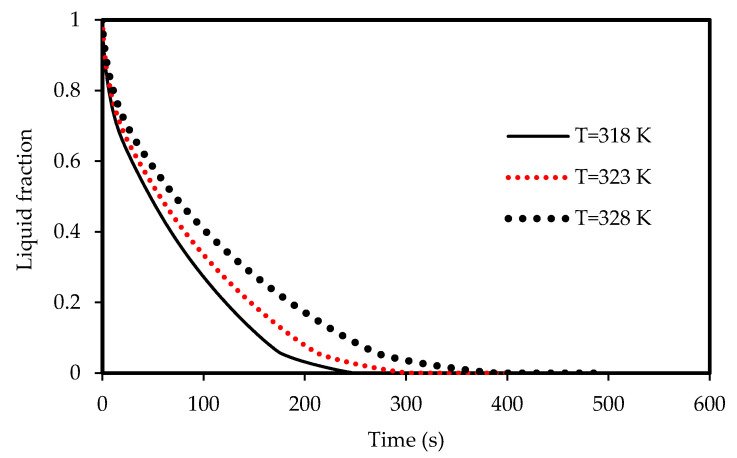
Evolution of the liquid-fraction for the solidification process in the horizontal corrugated channel for different airflow temperatures.

**Figure 24 molecules-25-04643-f024:**
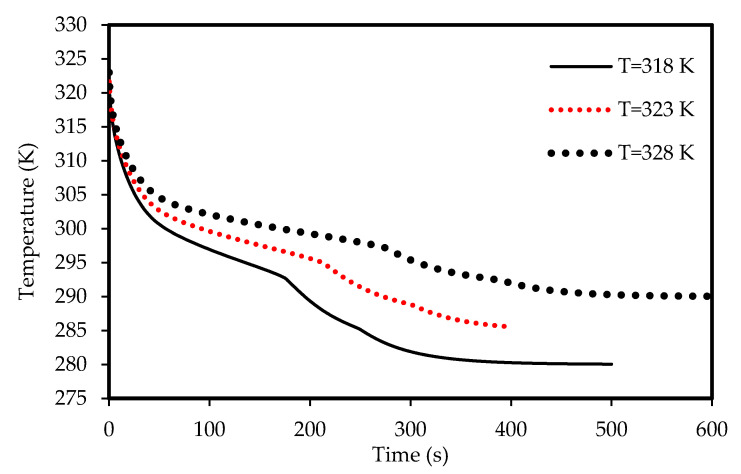
Variation of the temperatures for the solidification process in the horizontal corrugated channel for different airflow temperatures.

**Figure 25 molecules-25-04643-f025:**
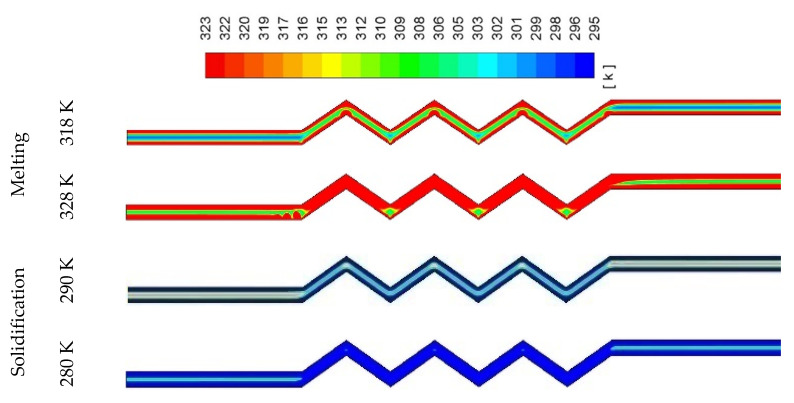
Contours of the temperature in the melting and solidification processes for different airflow temperatures after 200 s.

**Table 1 molecules-25-04643-t001:** Geometrical details of the zigzag mist eliminator.

L	Bend Angle (α)	Wavelength (λ)	Width (w)	Number of Bends (n)	Spacing (s)
100 mm	33. 1°	50 mm	300 mm	7	8 mm

**Table 2 molecules-25-04643-t002:** Thermo-physical properties of RT35 [60].

Property	ρ	$L_{f}$	$C_{p}$	k	μ	Liquidus Temperature (K)	Solidus Temperature (K)	β
Values	815	170	2.0	0.2	0.023	309	302	0.0006

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
