# Peer review of "Phase Change Process in a Zigzag Plate Latent Heat Storage System during Melting and Solidification"

_molecules, 2020, doi:10.3390/molecules25204643_

Round 1

Reviewer 1 Report

This article reported a 2D numerical work on the solid-liquid phase-change process in a zigzag plate latent heat storage system. The effects of the channel orientation, width, and HTF temperature on the melting and solidification processes are investigated. Generally, it is suitable to be published in Molecules. However, the following issues should be carefully addressed.

1) The authors should specifically highlight the novelty of the current work as well as the advantages of using zigzag structures.

2) Why the melting and solidification processes are both investigated in this study? In other words, this zigzag structure has any different effect on these two processes?

3) The literature review is not sufficient. There have been several novel studies on the role of fin structures, such as the tree-like fins on the solid-liquid phase-change processes. It is interesting to give a brief review of this.

4) The authors check the whole manuscript throughout since I have found many grammatical mistakes.

Author Response

Reviewer #1:

This article reported a 2D numerical work on the solid-liquid phase-change process in a zigzag plate latent heat storage system. The effects of the channel orientation, width, and HTF temperature on the melting and solidification processes are investigated. Generally, it is suitable to be published in Molecules. However, the following issues should be carefully addressed.

  • The authors should specifically highlight the novelty of the current work as well as the advantages of using zigzag structures.

Answer/ The novel aspect of this work is to study both the melting and solidification processes of the PCM numerically in a zigzag system under the effects of different width of the corrugated system and different temperature of heat transfer fluid. This sentence added to the last paragraph of the introduction (marked in red). And the advantage of the zigzag configuration is also mentioned in the last paragraph of the introduction too (marked in red).

  • Why the melting and solidification processes are both investigated in this study? In other words, this zigzag structure has any different effect on these two processes?

Answer/ Both the melting and solidification processes are studied in this work for two reasons, firstly, practically, all the heat storage system undergoes both processes to finish one cycle of work, so the author realised that studying both the processes under the effect of the same parameters is more sensible and convincing to present and analysed. Secondly, to expand the study and presenting the novel information and details regarding the effect of the studied parameters on the system in one work rather than two.

  • The literature review is not sufficient. There have been several novel studies on the role of fin structures, such as the tree-like fins on the solid-liquid phase-change processes. It is interesting to give a brief review of this.

Answer/ thanks to the reviewer for this suggestion, some studied have been added to the introduction (marked in red) regarding this aspect.

4) The authors check the whole manuscript throughout since I have found many grammatical mistakes.

Answer/ the whole manuscript has been checked grammatically.

Reviewer 2 Report

The article "Phase change process in a zigzag plate latent heat storage system during melting and solidification" focuses on a meaningful topic. However, several issues need major revisions.

The abstract should introduce the research topic explaining the novelty and the contribution of the presented work to the current knowledge. Quantitative data are necessary for the presentation of results (lines 29 -- 33).

In the introduction, the authors should add (with references) a short description of the "few existent studies on PCM zigzag-plate" (lines 122) and explain why their work is different and what is the novelty of their approach.

Figure 1: what mesures "w"?

Air velocity: The authors declare to use a "high" airflow rate for their simulations (line 142) without any quantitative data. Further, they use a laminar model. How the authors support both these assumptions? 

At lines 226-229, the authors declare that the corrugated region presents a higher heat-transfer rate due to the more extended heat exchange surface. Does the corrugated region increase the pressure losses inducing the formation of vortexes that increase the heat-transfer rate by convection? How could the author include this effect in their case studies if they neglect the fluid turbulence?

The orientation of the domain in space is not clear. Add a reference system in Figure1.

The validation of the model is too qualitative. Add quantitative data to demonstrate the grid independence (e.g., see Figure5 in Mirsandi, H., et al. "Bubble formation from an orifice in liquid cross-flow." Chemical Engineering Journal 386 (2020): 120902) and calculate the error and the accuracy of your model when compared with the literature [54]. 

The description of different plate configurations and operating conditions are confusing (lines 201-202; lines 279-281; lines 217-219). Include a concise description of the domain and the operating conditions (e.g., the PCM volume) of all simulations cases (I suggest to move in a separate section before results).

The authors should explain why they do not consider the effect of airflow velocity on the heat transfer rate and limit the study to the air temperature.    

Conclusions need more quantitative results, especially when comparing different case studies. Further, the authors should add more details on how their research contributes to the current design practice. 

Author Response

Reviewer #2:

The article "Phase change process in a zigzag plate latent heat storage system during melting and solidification" focuses on a meaningful topic. However, several issues need major revisions.

(1) The abstract should introduce the research topic explaining the novelty and the contribution of the presented work to the current knowledge. Quantitative data are necessary for the presentation of results (lines 29 -- 33).

Answer/ A new relevant opening has been added to the abstract on (lines 25 -- 27).

“Applying a well-performing heat exchanger is an efficient way to fortify the relatively low thermal response of phase-change materials (PCMs), which have broad application prospects in the fields of thermal management and energy storage”.

Also, the necessary qualitative data has been added on (lines 33 -- 35).

“It is found that the melting rate is about 8% faster, and the average temperature is 4 K higher in the corrugated region compared to the smooth region because of the large heat-exchange surface area in the corrugated region, which facilitates the faster rate of heat transport into the PCM channel”.

(2) In the introduction, the authors should add (with references) a short description of the "few existent studies on PCM zigzag-plate" (lines 122) and explain why their work is different and what is the novelty of their approach.

Answer/ Two additional paragraphs have been added to better respond to this reviewer’s comment on (lines 117 -- 145).

“Increasing the surface area for heat transfer via fin or tree fin enhances the melting and solidification rates of the PCM system. Researchers studied the fin complained [5, 17,28,31,37, 51] and tree finned complained [48-50]. Sathe and Dhoble [51] analyzed the dynamic thermal efficiency of PCM charging in an inclined finned rectangular enclosure with the top surface heating source. They found that the charging rate does not significantly be affected by combining extra fins (more than three), although it considerably enhances the PCM thermal efficiency. Yeldiz et .al [49] numerically studied the thermal convection inside a PCM enclosure combined with two types of fins as rectangular and tree-shape branched fins and various length-to-height ratio of the rectangular fin. They found that the rectangular fin is more effective than the tree-shape fin with the same mass. Yu et al. [50] examined the potential of a fractal tree-like fin to improve the melting property of PCM in latent TES units. It was found that the good distribution of tree-like fins can help to develop better melting acceleration due to the more uniform temperature nature of tree-like fins compared to that of the corresponding plate fins.

Interruptions in the heat-exchange surface like zigzag configuration can bring a significant improvement on the melting and solidification rates, thereby contributing to a superior storage performance of the PCM-based TES system. This type of surface interruption is preferable since the zigzag configuration enables the PCM in charge to have larger heat-exchange surface area compared to the conventional flat-plate heat exchanger. This makes the heat energy absorption and release to be higher, thus melting and solidification of PCM turn to proceed at faster rates. As per literature, very few studies up to now have been conducted to explore the use of zigzag-plate heat exchanger as a good performing design for PCM containment. Wang et al. [52] were the first investigated the effect multiple-PCM arrangement within a zigzag plate TES unit. They recognized that a potential melting improvement can be achieved if a larger temperature difference between the PCM layers is applied. In their other work [53], they studied the solidification of multiple PCMs within a zigzag plate type containment. They stated that the zigzag containment brings a very positive impact on the temperature distribution and the overall solidification response of the multiple-PCM module.

In this study, the energy storage potential and thermofluidic behavior of a PCM within a zigzag plate TES system under the effects of different widths of the corrugated surface and different temperatures of the heat transfer fluid. Therefore, the main objective of this study is to explore through numerical simulation the melting and solidification mechanisms of PCM in a zigzag-plate heat exchanger being applicable to use with compact air-conditioners. Different zigzag-plate configurations are investigated to determine the optimized configuration under different heat load conditions. Different orientations of the heat exchanger including the horizontal and vertical arrangements are considered. The effects of the channel’s width as well as the wall temperature of the heat exchanger are studied in terms of the temperature distribution, liquid-fraction contours, average-temperature curves, and liquid-fraction profile. Findings based on this study would contribute to the ongoing improvement efforts in the design, analysis and operation of TES systems for realistic applications”.

(3) Figure 1: what mesures "w"?

 Answer/ The measure “w” has been added to Figure 1 of the revised manuscript.

(4) Air velocity: The authors declare to use a "high" airflow rate for their simulations (line 142) without any quantitative data. Further, they use a laminar model. How the authors support both these assumptions? 

Answer/ This is a typo error. It is now corrected on lines 161-163 as “It is assumed that the flow rate of the HTF is moderate enough to have an almost constant temperature at the walls of the PCM section”.

(5) At lines 226-229, the authors declare that the corrugated region presents a higher heat-transfer rate due to the more extended heat exchange surface. Does the corrugated region increase the pressure losses inducing the formation of vortexes that increase the heat-transfer rate by convection? How could the author include this effect in their case studies if they neglect the fluid turbulence?

Answer/ The following explanation is added on lines 259-262 to better respond to this reviewer’s comment.

“It would be worthy to remind the reader here that the flow of melted PCM during both modes of melting and solidification is buoyancy-driven and the velocity remains too low to form fluid turbulence or vortexes. The velocities of melted PCM in all cases within the scope of this study are in the order of 10-4 m/s”.  

(6) The orientation of the domain in space is not clear. Add a reference system in Figure1.

 Answer/ The orientation of the domain in figure 1 is now modified in the revised manuscript.

(7) The validation of the model is too qualitative. Add quantitative data to demonstrate the grid independence (e.g., see Figure5 in Mirsandi, H., et al. "Bubble formation from an orifice in liquid cross-flow." Chemical Engineering Journal 386 (2020): 120902) and calculate the error and the accuracy of your model when compared with the literature [54]. 

Answer/ Thanks for the comment. The results of grid independence analysis is added and presented in a new figure as follows:

A series of numerical simulations with different cell numbers of 172016, 267002, and 419903 proves how generously the mesh was generated to have a grid-independent solution. The variation of the liquid fraction with different numbers of cells are presented in Figure 3 (a). As shown, the results for all grid resolutions are identical with the maximum difference of less than 1.5 %. Therefore, the mesh with the cell number of 172016 is sufficient for the modelling. These results prove that the solution is mesh-independent and reliable enough to be verified against other studies’ results. 

For the validation, the error and deviation are also added as follows:

In the temperature profile, the maximum deviation between the present numerical study and experimental data is almost 9% while it is around 5% compared with the numerical study of Mat et al. For the liquid fraction, the maximum deviation is 4% between the present numerical results and the numerical study of Mat et al.

(8) The description of different plate configurations and operating conditions are confusing (lines 201-202; lines 279-281; lines 217-219). Include a concise description of the domain and the operating conditions (e.g., the PCM volume) of all simulations cases (I suggest to move in a separate section before results).

Answer/ Description of the different plate configurations has been modified at different places in the revised manuscript as follows:

On lines (224-231), this statement has been added, “Three different cases are considered for the plate TES system, namely: the horizontal corrugated plate, vertical corrugated plate, and horizontal non-corrugated plate. The transient melting evolution for the three studied cases is demonstrated in figure 4. It can be observed from this figure that the horizontal corrugated plate presents the highest melting performance, even it has a delicate difference compared with the vertical case. However, the difference is more noticeable compared with the horizontal flat channel. For the same channel width (w), the total times that the PCM take for complete melting is about 315 s in a corrugate case against 343 s in the smooth plate case so that the time saving due the use of the corrugate channel is about 8%”.

On lines (241-246), this statement has been added, “The PCM solidification rate for the case of the corrugated plate preserves to be quicker than that of the smooth flat plate over the entire solidification process as shown in figure 6. This is due to the larger surface area of the corrugated surface configuration. For the same size of PCM in both cases, the total times that the PCM would take for complete solidifying is 280 s in the case of the corrugate plate against 303 s in the case of the smooth plate. Therefore, the time saving due to the use of the corrugate channel is about 7% during the solidification mode”.

On lines (292-302), this statement has been added, “The effects of the channel width are illustrated in figures (11 - 14) for both melting and solidification modes. The channel of 0.5 W is thinner than the other two cases, thus holds less amount of PCM as shown in Figure 11. This helps the PCM to completely melt within 94s faster than the reference case of width (W=0.9cm) and the case of double width (2 W) width, which melt in 332s and 776s, respectively. This implies that reducing the zigzag plate width to 0.5 W can save the melting rate by 72 % compared to that of the reference case of W=0.9cm. This behavior is due to transferring the same amount of heat through the same surface area to half the volume of PCM. Figure 12 compares the mean PCM temperature of the thinner case (0.5 W) to the other two cases of the reference width (W=0.9cm) and the double width (2 W). For the same reasons mentioned above, the thinner case reaches to the higher melting temperature within shorter duration compared to the other two cases under the consideration”.

(9) The authors should explain why they do not consider the effect of airflow velocity on the heat transfer rate and limit the study to the air temperature.    

 Answer/ In order not to make the problem unwieldy, there is the need to limit the number of scenarios studied. Here, the effort was mainly devoted to exploring how the air temperature with the proposed zigzag configuration can improve the heat transfer rate during the PCM melting and solidification. Expanding the research to investigate the airflow velocity is a logical next step in our ongoing research on the performance improvement of PCM-based TES systems.

(10) Conclusions need more quantitative results, especially when comparing different case studies. Further, the authors should add more details on how their research contributes to the current design practice. 

To respond to this reviewer’s comment, the following text has been on lines 390-395 of the revised manuscript.

“The total time saving due to the use of the corrugate channel is about 8% and 7% during the melting and solidifying mode, respectively. Moreover, the PCM melting and solidifying rates are faster in the thinner channel as the system holds less PCM mass. It is found that reducing the zigzag plate width from 2 W (W=0.9 cm) to 0.5 W can improve the melting rate by 8 and 12 times faster than the 2W, respectively”.

And to show on how the research could contribute to the current knowledge, the following is included on lines 399-402.

“These results may provide an insight into the possible optimization and improvement of the PCM containment design using a zigzag configuration to provide better control of the system size and the traditional thermal energy losses”.

Round 2

Reviewer 1 Report

Accept as it is.

Author Response

Thanks the reviewer for reviewing the paper

Reviewer 2 Report

Based on this answer of the authors from last revision's round:

"In order not to make the problem unwieldy, there is the need to limit the number of scenarios studied. Here, the effort was mainly devoted to exploring how the air temperature with the proposed zigzag configuration can improve the heat transfer rate during the PCM melting and solidification. Expanding the research to investigate the airflow velocity is a logical next step in our ongoing research on the performance improvement of PCM-based TES systems."

The authors should add in the conclusions the limits of their work and future developments.

I strongly recommend checking the language and improve the writing style of the manuscript paying particular attention to the clarity of the sentences (reduce their length and the use of the preposition.)

Author Response

Thanks to the reviewer for reviewing the paper. Please find the answers in the following:

  1. "In order not to make the problem unwieldy, there is the need to limit the number of scenarios studied. Here, the effort was mainly devoted to exploring how the air temperature with the proposed zigzag configuration can improve the heat transfer rate during the PCM melting and solidification. Expanding the research to investigate the airflow velocity is a logical next step in our ongoing research on the performance improvement of PCM-based TES systems."

Answer: The main focus of this study is on the effects of the corrugated plate heat exchanger during both melting and solidification compared with the straight channel for different dimensions of the heat exchanger. Therefore, as mentioned inline 161, it is assumed that the flow rate of the HTF is moderate enough to have an almost constant temperature at the walls of the PCM section and as a result, the constant temperature boundary condition is used for the walls without modelling the airflow. In other words, it was assumed that the channel is isothermally cooled or warmed by the HTF flowing at a constant temperature. This also enables the heat to only transfer to or from the PCM to realize only the effects of the corrugated plate on the phase change process. Furthermore, the effects of the HTF temperature was investigated comprehensively as the only effective parameters of the HTF. For the effect of HTF Reynolds number, it was proved that by increasing the Reynolds number, the rate of heat transfer from the HTF to the PCM increases due to the higher convection heat transfer which has been widely established in the literature and therefore it is not investigated quantitatively in this study [1-4]. Furthermore, as mentioned, for high flow rates of the HTF, the assumption of considering constant wall temperature is meaningful [5] and increasing the velocity has negligible effect as mentioned in different papers in the literature [5-7]. This explanation related to the effect of Reynolds number is added on page 4.

  1. The authors should add in the conclusions the limits of their work and future developments.

Answer: Thanks to the reviewer. In future studies, we are going to increase the number of zigzags by changing the angles of the corrugated plate in a constant length. The HTF is going to be modelled completely and the effect of different configurations of the plates is going to be investigated. From the numerical point of view, there is no limitation and there are some meaningful assumptions that make solving the problem easier. The assumptions of constant temperature for the walls is modified in future studies to have more accurate results. This explanation is added at the end of the conclusion according to the reviewer comments.

  1. I strongly recommend checking the language and improve the writing style of the manuscript paying particular attention to the clarity of the sentences (reduce their length and the use of the preposition.)

Answer: Thanks to the reviewer. The paper is double-checked completely and the language is checked and improved. Long sentences are reduced to smaller ones and the use of preposition is double-checked.

References

[1] R. Elbahjaoui, H. El Qarnia, Transient behavior analysis of the melting of nanoparticle-enhanced phase change material inside a rectangular latent heat storage unit, Applied Thermal Engineering, 112 (2016).

[2] A. Shahsavar, H.M. Ali, R.B. Mahani, P. Talebizadehsardari, Numerical study of melting and solidification in a wavy double‑pipe latent heat thermal energy storage system, Thermal analysis and calorimetry, (2020).

[3] A. Shahsavar, J. Khosravi, H.I. Mohammed, P. Talebizadehsardari, Performance evaluation of melting/solidification mechanism in a variable wave-length wavy channel double-tube latent heat storage system, Journal of Energy Storage, 27 (2020) 101063.

[4] A. Shahsavar, A. Shaham, P. Talebizadehsardari, Wavy channels triple-tube LHS unit with sinusoidal variable wavelength in charging/discharging mechanism, International Communications in Heat and Mass Transfer, 107 (2019) 93-105.

[5] Z. Liu, Y. Yao, H. Wu, Numerical modeling for solid–liquid phase change phenomena in porous media: Shell-and-tube type latent heat thermal energy storage, Applied Energy, 112 (2013) 1222-1232.

[6] J.M. Mahdi, S. Lohrasbi, E.C. Nsofor, Hybrid heat transfer enhancement for latent-heat thermal energy storage systems: A review, International Journal of Heat and Mass Transfer, 137 (2019) 630-649.

[7] J.M. Mahdi, H.I. Mohammed, E.T. Hashim, P. Talebizadehsardari, E.C. Nsofor, Solidification enhancement with multiple PCMs, cascaded metal foam and nanoparticles in the shell-and-tube energy storage system, Applied Energy, 257 (2020) 113993.